# Nutrient and arsenic biogeochemistry of *Sargassum* in the western Atlantic

**Dennis Joseph McGillicuddy Jr.** [1] ✉, **Peter Lynn Morton**[2], **Rachel Aileen Brewton**[3], **Chuanmin Hu**[4], **Thomas Bryce Kelly**[5], **Andrew Robert Solow**[1] & **Brian Edward Lapointe**[3]

The oceanographic ecology of pelagic *Sargassum*, and the means by which these floating macroalgae thrive in the nutrient-poor waters of the open ocean, have been studied for decades. Beginning in 2011, the Great Atlantic *Sargassum* Belt (GASB) emerged, with *Sargassum* proliferating in the tropical Atlantic and Caribbean where it had not previously been abundant. Here we show that the nutritional status of *Sargassum* in the GASB is distinct, with higher nitrogen and phosphorus content than populations residing in its Sargasso Sea habitat. Moreover, we find that variations in arsenic content of *Sargassum* reflect phosphorus limitation, following a hyperbolic relationship predicted from Michaelis-Menten nutrient uptake kinetics. Although the sources of nutrients fueling the GASB are not yet clear, our results suggest that nitrogen and phosphorus content of *Sargassum*, together with its isotopic composition, can be used to identify those sources, whether they be atmospheric, oceanic, or riverine in origin.

The floating macroalgae *Sargassum* spp. serves as a habitat for more than 200 types of organisms, including 10 endemic species[1–3], and provides habitat for fish nursery, spawning, and foraging[4,5]. Neonate and juvenile sea turtles use *Sargassum* habitat for feeding as well as protection from predators[6,7], and several species of seabirds forage over *Sargassum*[8,9]. In contrast to these traditionally beneficial ecological impacts, the Great Atlantic *Sargassum* Belt[10] (GASB) has caused unprecedented inundations of *Sargassum* on Caribbean and Florida coastlines with deleterious effects on near-shore seagrass and coral reef ecosystems[11–14], and declines in turtle hatchling survival[15]. The GASB has also presented challenges to regional economies, particularly those that rely heavily on tourism[16,17]. *Sargassum* decay on coastal margins can cause respiratory and other human health issues[18], and the presence of arsenic in *Sargassum* tissue[19–21] puts significant constraints on the utilization of the biomass that washes ashore[22–24].

A complex set of interconnected hypotheses has been offered to explain the dynamics of the GASB[10,25,26]. Anomalous wind patterns in 2009-2010 are thought to have introduced *Sargassum* from its habitat in the Sargasso Sea into the eastern North Atlantic where it was subsequently entrained into the equatorial current system[27]. Since that time, *Sargassum* has increased dramatically in the tropical Atlantic, begging the question of nutrient supply. A variety of nutrient sources have been suggested[10,27–29], including upwelling, vertical mixing, discharge from the Amazon and Congo rivers, and atmospheric deposition. However, the causes of the GASB and the mechanisms controlling its seasonal to interannual variability remain unknown. An underlying question is: is this massive abundance of *Sargassum* a result of higher nutrient availability in the GASB?

In this work we show that GASB *Sargassum* populations are enriched in both nitrogen and phosphorus content relative to its Sargasso Sea habitat, clearly identifying nutrient supply as a primary driver of this phenomenon. In addition, we demonstrate that *Sargassum*'s arsenic content varies with the degree of phosphorus limitation, linking our observations with a theoretical prediction based on nutrient uptake kinetics. Our results show that ascertaining the nutrient sources and their regulation is essential to understand the underlying causes of this basin-scale phenomenon—and only then will society

[1]Woods Hole Oceanographic Institution, Woods Hole, MA, USA. [2]Department of Oceanography, Texas A&M University, College Station, TX, USA. [3]Harbor Branch Oceanographic Institute, Florida Atlantic University, Fort Pierce, FL, USA. [4]College of Marine Science, University of South Florida, St. Petersburg, FL, USA. [5]College of Fisheries and Ocean Science, University of Alaska Fairbanks, Fairbanks, AK, USA. ✉e-mail: dmcgillicuddy@whoi.edu

have a conceptual basis on which to design potential strategies to mitigate the consequences of the GASB.

## Results

### *Sargassum* distributions and nutritional status

In order to address this question, *Sargassum* tissue samples were collected in spring 2021 along with hydrographic and nutrient measurements on two hydrographic sections in the western Atlantic (Methods) that intersected the western portion of the GASB (Fig. 1). The samples were separated into two species *S. fluitans* and *S. natans* based on their morphology[30]. *S. fluitans* and *S. natans* co-occurred in the southern portions of both A20 and A22. *S. natans* was more prevalent north of 24°N on A22, and north of 27°N on A20. Both species were present at 32°N on A20 as well as the farthest north sample, collected just south of the Gulf Stream on A22.

*Sargassum* elemental composition was similar between species (Fig. 2, Supplementary Table 1), a finding that is consistent with earlier studies[29]. Our data reveal three distinct regimes within the sampling domain. In the Sargasso Sea, carbon content tends to be relatively high (particularly in the eastern transect A20), whereas nitrogen and phosphorus content tend to be low. Nutritional status in the GASB was much better, with the nitrogen and phosphorus content of *Sargassum* in the Caribbean and western tropical Atlantic significantly higher than in the Sargasso Sea (Supplementary Fig. 1, Supplementary Table 2). Interestingly, nitrogen and phosphorus content were highest in the northern Sargasso Sea.

These trends are also evident in elemental ratios (Supplementary Fig. 2). As expected, C:N was high in the Sargasso Sea, driven by higher carbon content and lower nitrogen content (Fig. 2). These ratios are higher in A20 than in A22, particularly in *S. natans*. This zonal gradient with C:N increasing to the east is driven by both increasing carbon content and decreasing nitrogen content. C:P is also high in the Sargasso Sea, although the zonal gradient evident in C:N is not clear in C:P. N:P shows a pronounced maximum in the Sargasso Sea region of A22, which is driven by a combination of increased N content and decreased P content. Little meridional trend in N:P is evident in A20, and in aggregate N:P in the Sargasso Sea differs less from the surrounding regions than does C:N and C:P (Supplementary Fig. 1).

Based on the seasonal analysis conducted on prior data sets[29], the springtime samples described herein would be expected to be at their seasonal maxima of nitrogen and phosphorus content, and minima of their C:N and C:P ratios.

### Nutrient sources

Because *Sargassum* is subject to both surface currents and wind, hydrodynamic transport plays a key role in its dynamics[31–33]. In order to assess the recent history of our *Sargassum* samples, passive particles were inserted into a numerical ocean model hindcast at each collection site and tracked backward in time (Fig. 3; see Methods). The southernmost sample of A20 appears to have come from the southeast, whereas samples in the 10–30°N latitude band come primarily from the east. Exceptions to that pattern include an eddy-like flows from 25°N to 28°N on A22. Samples collected north of 30°N on A20 appear to have come from the north, consistent with a Gulf Stream origin. The northernmost sample on A22 was clearly under the influence of the Gulf Stream, although its point of entry into the Gulf Stream varied

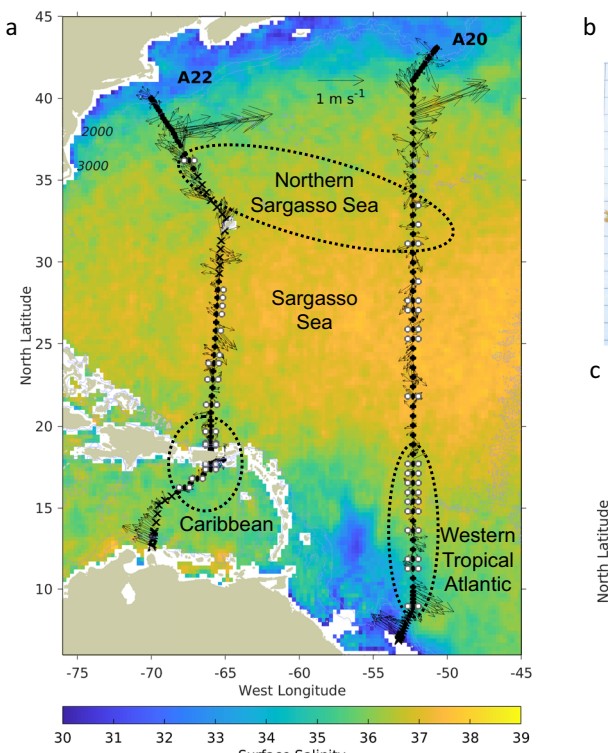

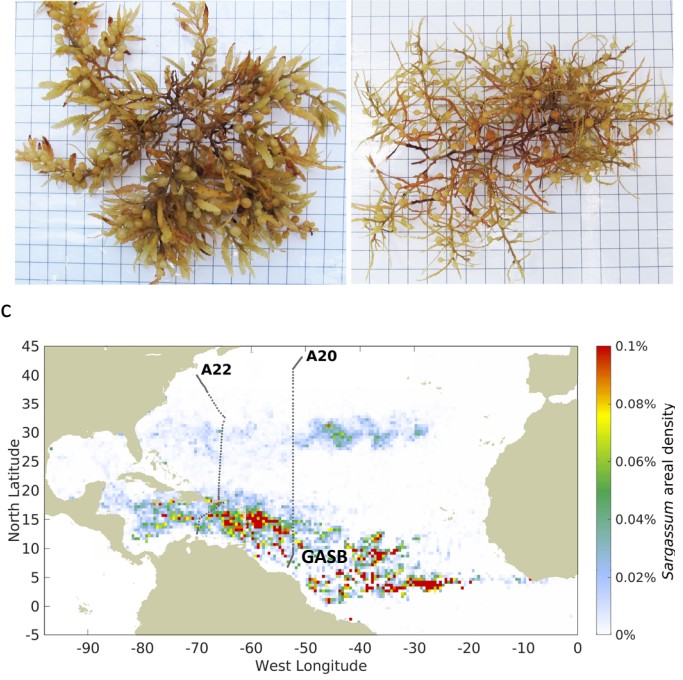

**Fig. 1 | *Sargassum* sampling on GO-SHIP lines A20 (16 March–16 April 2021) and A22 (20 April–16 May 2021). a** White circles indicate the collection of *Sargassum* samples, with *S. fluitans III* shown to the left and *S. natans I* to the right of the station locations, which are indicated as black circles (where *Sargassum* sampling was possible), and black Xs (where *Sargassum* sampling was not possible). Dashed black lines indicate sample groupings in the northern Sargasso Sea, Sargasso Sea, western tropical Atlantic, and Caribbean. Velocity vectors are from the uppermost bin (centered at 29 m depth) of the ship's acoustic doppler current profiler. The surface salinity field is comprised of a time-average of Soil Moisture and Ocean Salinity (SMOS) satellite measurements for the cruise period (16 March–16 May, 2021). **b** Photographs of *S. fluitans III* and *S. natans I* samples on a 1 cm grid background. Photo credit: Amy Siuda and Jeffrey Schell, Sea Education Association. **c** Location of GO-SHIP lines A20 and A22 relative to the Great Atlantic *Sargassum* Belt (contiguous area of *Sargassum* coverage surrounding the annotation) for the cruise period estimated from MODIS data. Blue lines are the 2000 m and 3000 m isobaths.

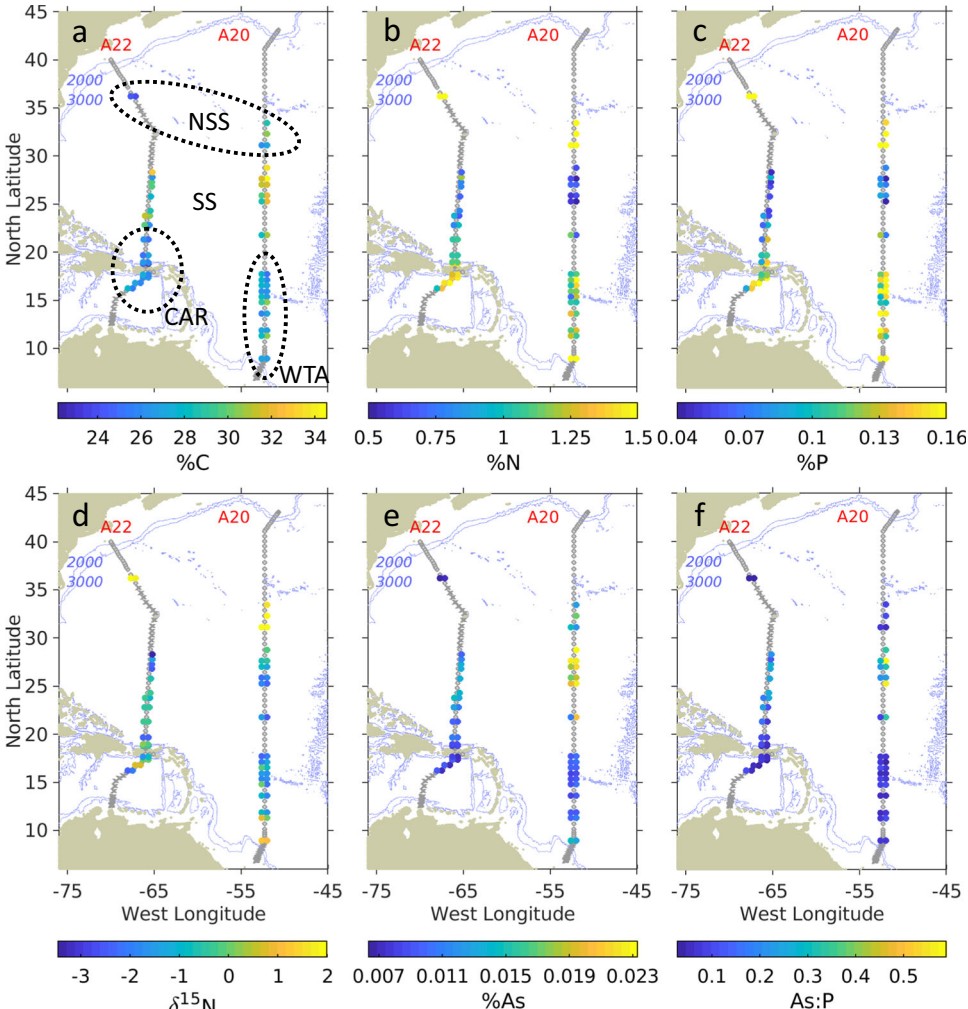

**Fig. 2 | *Sargassum* tissue data from GO-SHIP lines A20 and A22, as depicted in Fig. 1. a** Carbon content (% dry weight). **b** Nitrogen content. **c** Phosphorus content. **d** $\delta^{15}N$. **e** Arsenic content. **f** Arsenic to phosphorus ratio. *S. fluitans III* is shown to the left of transects A20 and A22, and *S. natans I* to the right. Stations, where no *Sargassum* was found, are shown as circles, and stations where *Sargassum* sampling was not possible are shown as Xs. Dashed black lines in the upper left panel indicate sample groupings in Northern Sargasso Sea (NSS), Sargasso Sea (SS), Caribbean (CAR), and Western Tropical Atlantic (WTA), as in Fig. 1. Blue lines are the 2000 m and 3000 m isobaths.

widely within the ensemble—ranging from the continental shelf north of Cape Hatteras to the South Atlantic Bight, with some in the core of the Gulf Stream as far south as 30°N.

This information, together with $\delta^{15}N$ values of the *Sargassum* tissue, facilitates some inferences about nitrogen sources. Specifically, *Sargassum* utilizing riverine nitrogen tend to have enriched (positive) $\delta^{15}N$ values, whereas those fueled by near-surface oceanic and atmospheric sources have lower (negative to slightly positive) $\delta^{15}N$ values[29]. Nitrogen fixation typically results in a fractionation of −2‰[34].

Patterns in $\delta^{15}N$ from our survey reveal spatial coherence, with no systematic difference between *S. fluitans* and *S. natans* (Fig. 2). Depleted $\delta^{15}N$ values in the Sargasso Sea, Caribbean, and western tropical Atlantic could result from nitrogen fixation by epiphytic cyanobacteria[35,36] or perhaps atmospheric deposition[37]. The highest $\delta^{15}N$ values occurred in the northern Sargasso Sea, with values of approximately +2‰ potentially reflecting riverine sources along the U.S. east coast (as indicated by spatial connectivity depicted in Fig. 3), or as distant as the Gulf of Mexico which has been identified as a source region for *Sargassum* in the western Atlantic[38]. For example, a coherent plume emanating from the Mississippi River in the summer of 2004 was entrained into the Loop Current, reaching the Straits of Florida in approximately 25 days, continuing northward in the Gulf Stream to be

found in the South Atlantic Bight another 25 days after that[39]. Our northern Sargasso Sea samples were collected another 600 nm farther downstream, so assuming a flow of 4 kt, they could have been influenced by Mississippi River water discharged as recently as 150 days before sampling. With end-member $\delta^{15}N$ values of +6−8‰ in the northern Gulf of Mexico *Sargassum*[29], dilution to +2‰ would require several doublings of the population using oceanic sources of nitrogen −which is certainly possible based on observed growth rates[36,40,41]. Alternatively, enrichment of $\delta^{15}N$ in samples from the northern Sargasso Sea could be caused by upwelling and/or vertical mixing, as values of +2‰ are characteristic of nitrate in the upper thermocline in that region[42,43].

Enriched $\delta^{15}N$ values were also found in the southernmost samples of the western tropical Atlantic (Fig. 2), under the direct influence of the Amazon River plume (as evidenced by the salinity distribution in Fig. 1) which has been implicated as a nutrient source for the GASB[10,27−29,44]. There was one station in the Caribbean interior (17°N, 67°W) where similarly high $\delta^{15}N$ values were measured, but the corresponding salinity (Fig. 1) does not bespeak riverine influence. Particle backtracking calculations suggest source waters to the east-southeast (Fig. 3), and given the cross-isohaline transport enabled by wind drag, Amazon River influence is plausible. However, it is noteworthy that

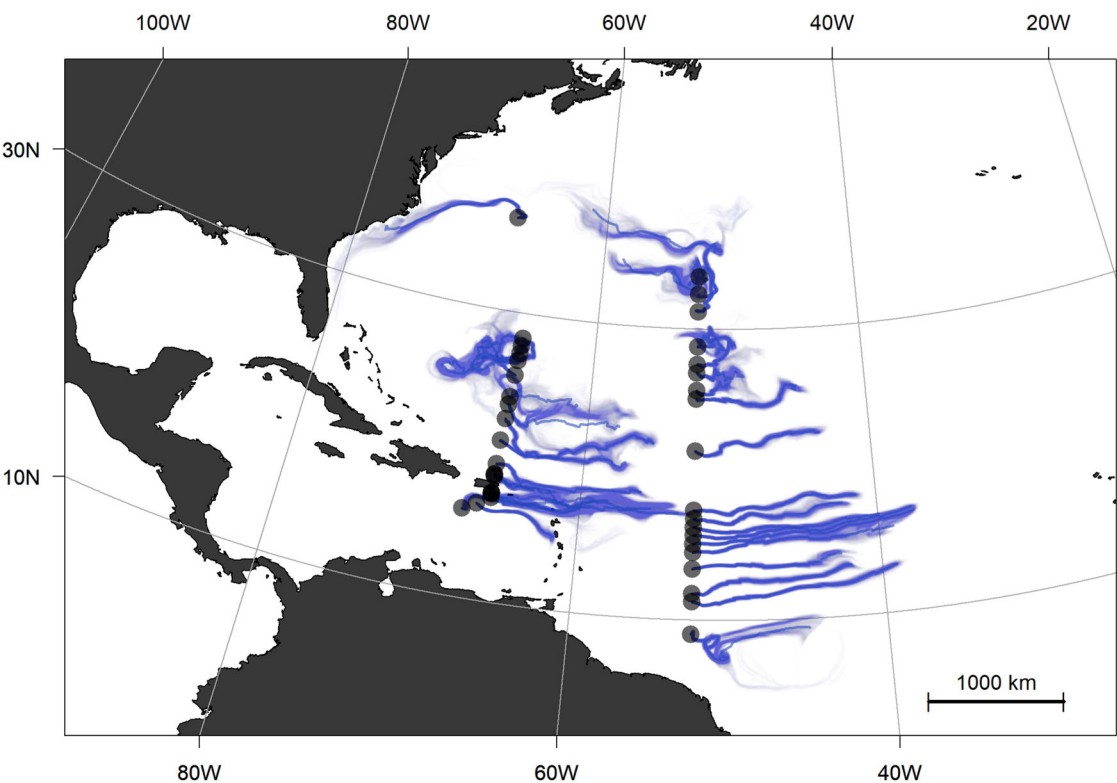

**Fig. 3 | Origins of *Sargassum* spp. samples from GO-SHIP lines A20 and A22.** Surface particles deployed at sample collection sites (Figs. 1, 2) were tracked back in time for 60 days. The centroid of the ensemble is shown as a bold blue line with individual trajectories shown in lighter blue.

neither the elemental composition (Fig. 2) nor nutrient ratios (Supplementary Fig. 2) in the high-δ¹⁵N samples of the tropical Atlantic and Caribbean Sea show corresponding anomalies.

Hydrographic data from sections A20 and A22 provide environmental information with which to interpret the *Sargassum* observations. The nitracline and phosphocline are deepest in the warm and salty waters of the Sargasso Sea, and shallower in the warmer and fresher waters of the Caribbean and western tropical Atlantic (Supplementary Fig. 3). Both nutriclines are shallower in the colder and fresher waters of the northern Sargasso Sea. Near-surface nutrient concentrations are generally very low except for the northernmost portions of each transect where the nutriclines outcrop. Nitrate and phosphate are at or below the limit of detection in the surface waters Sargasso Sea (Supplementary Fig. 4). Nitrate is enhanced in the Caribbean, and less so in the western tropical Atlantic, where surface phosphate is elevated. The presence of detectable nitrate and phosphate in these areas is consistent with the enhanced nutritional status of *Sargassum* in the GASB.

### Arsenic biogeochemistry

The fact that *Sargassum* bioaccumulates arsenic has generated considerable interest, particularly because the associated toxicity puts practical limitations on the valorization of the biomass inundating coastal communities of the GASB. Our survey revealed regional scale variations in arsenic content (Fig. 2). Arsenic content was lowest in the GASB (Caribbean and Western Tropical Atlantic samples) and highest in the Sargasso Sea, particularly in the eastern transect A20. Other stations in A20 show modest enrichment, such as in the southernmost samples at 9°N, 52°W and the Northern Sargasso Sea samples at 31–33°N, 52°W. There are no systematic differences in arsenic content between *S. fluitans* and *S. natans* (Supplementary Table 1), although interspecies variability is apparent at some stations (Fig. 2).

The high arsenic content of *Sargassum* in the subtropical gyre is accompanied by the lowest phosphorus content, which is consistent

with the depression of the phosphocline (Supplementary Fig. 3) and surface dissolved phosphate concentrations generally below the limit of detection (Supplementary Fig. 4). Thus, the arsenic to phosphorus ratio of *Sargassum* of the subtropical gyre stands out as uniquely high in that region (Fig. 2), where *Sargassum* is known to be phosphorus limited[45].

Uptake of arsenate ($AsO_4^{3-}$) by aquatic plants occurs as a byproduct of phosphorus uptake, owing to its chemical similarity with the phosphate ion ($PO_4^{3-}$)[46,47]. Using the standard Michaelis-Menten form, the uptake of dissolved phosphate and arsenate can be expressed as:

$$\rho_P = \mu_P \frac{\left[PO_4^{3-}\right]}{k_P + \left[PO_4^{3-}\right]} \text{ and } \rho_{As} = \mu_{As} \frac{\left[AsO_4^{3-}\right]}{k_{As} + \left[AsO_4^{3-}\right]} \quad (1)$$

For low concentrations of phosphate and arsenate, uptake is approximately linear in concentration:

$$\rho_P \approx \mu_P \frac{\left[PO_4^{3-}\right]}{k_P} \text{ and } \rho_{As} \approx \mu_{As} \frac{\left[AsO_4^{3-}\right]}{k_{As}} \quad (2)$$

The ratio of As to P uptake is therefore:

$$\frac{\rho_{As}}{\rho_P} = \frac{\mu_{As} k_P}{\mu_P k_{As}} \frac{\left[AsO_4^{3-}\right]}{\left[PO_4^{3-}\right]} \quad (3)$$

Surface arsenate concentrations in the tropical and subtropical Atlantic are relatively uniform in comparison with phosphate, and up to an order of magnitude lower[48]. Moreover, surface concentrations of arsenate and phosphate are uncorrelated[49], so the ratio of As to P

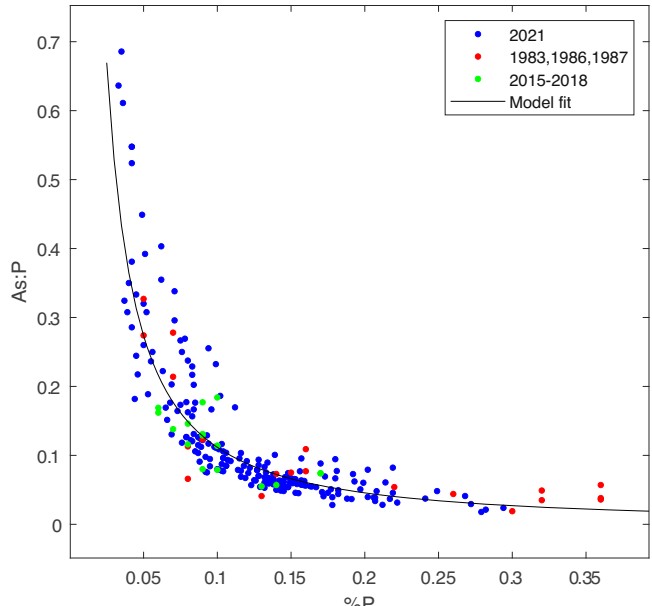

**Fig. 4 | Arsenic to phosphorus ratio as a function of phosphorus content in *Sargassum* tissue.** Data from GO-SHIP lines A20 and A22 in 2021 are in blue (*N* = 200), 1983–1987 in red (*N* = 20), and 2015–2018 in green (*N* = 21). Black line is the fit described in the Supplementary Note.

uptake can be simplified to:

$$\frac{\rho_{As}}{\rho_P} \sim \frac{1}{\left[PO_4^{3-}\right]} \qquad (4)$$

Assuming that the tissue content of these two constituents reflects their proportionate uptake, one would expect that the ratio of arsenic to phosphorus in *Sargassum* would be a hyperbolic function of phosphorus content:

$$\frac{\%As}{\%P} \sim \frac{1}{\%P} \qquad (5)$$

This prediction is qualitatively consistent with the present data as well as prior observations dating back to the 1980s (Fig. 4), although the dependence on phosphorus content varies as $\%P^{-1.3}$ rather than $\%P^{-1.0}$ predicted by the theory. The reason for this supra-hyperbolic dependence is not known, but it is statistically reliable with a *p*-value < 0.001 (Supplementary Note). This is a clear demonstration of arsenic content as a diagnostic of phosphorus limitation in a natural population of marine algae and is consistent with laboratory studies[50,51], biogeochemical proxies based on dissolved As speciation in the ocean[49], and other recent field data[52].

It could be argued that the As:P versus P relationship in Fig. 4 is a result of the intrinsic association among variables: for two random variables a and b, plotting the ratio a:b versus b will take on a hyperbolic form. However, the elemental composition of *Sargassum* is not random. For example, the As:C ratio does not show a hyperbolic dependence on carbon content (Supplementary Fig. 5). Interestingly, As:N varies hyperbolically with nitrogen content, which we attribute to covariation in nitrogen and phosphorus content.

Notwithstanding our theoretical prediction of the hyperbolic relationship of As:P and P content, one might also expect a negative correlation between As and P content as a symptom of phosphorus stress in *Sargassum*. Observations are also consistent with this expectation (Supplementary Fig. 6). A similar negative correlation is observed between arsenic and nitrogen content, which we again

attribute to the correlation between nitrogen and phosphorus content. In contrast, As content is positively correlated with carbon content.

## Discussion

Nutrient limitation of oceanic *Sargassum* populations in their native habitat was demonstrated decades ago[45,53], and enhanced nutrient availability has been advanced as a key factor in stimulating the GASB[10,27,29]. We show clearly for the first time that *Sargassum* in the GASB is enhanced in both nitrogen and phosphorus, indicative of a healthy and thriving population. Stable nitrogen isotope values point to riverine sources in some circumstances and are more equivocal in others. Distinguishing the various nutrient sources sustaining the GASB will require systematic snapshots of nutrient content and isotopic composition across its entire breadth. Presumably, the closer one gets to the source, the higher the nitrogen and / or phosphorus content of *Sargassum* should be. In that sense, basin-wide patterns in nitrogen and phosphorus elemental composition could provide the fingerprinting necessary to unequivocally determine the sources. However, given the strong seasonal to interannual variability intrinsic to the GASB, it will be essential that such snapshots be synoptic, which poses significant practical challenges for a phenomenon of this scale.

Our demonstration of arsenic content as an indicator of phosphorus stress in natural populations of *Sargassum* also has considerable implications. *Sargassum* inundating coastal areas of the GASB already contains arsenic concentrations that can exceed safe thresholds for consumption[22–24]. If the phosphorus supply to the GASB were to wane relative to that of nitrogen, our findings would suggest that the arsenic content of *Sargassum* in that area would rise even further, perhaps up to levels currently observed in the Sargasso Sea. This would have important implications for management if the nutrient sources turn out to be anthropogenic.

For all these reasons, expanded observational and modeling studies are needed to understand the GASB's physical, biological, and chemical drivers. Moreover, the societal need for scientific understanding is urgent: improved seasonal to interannual predictions would offer tremendous value for proactive planning and response, while quantification of the underlying causes could inform potential management actions to mitigate the problem.

## Methods

Sampling was conducted on R/V *Thomas G. Thompson* voyages TN389 (16 March–16 April 2021) and TN390 (20 April–16 May 2021), occupying GO-SHIP lines A20[54] and A22[55], respectively.

### Hydrography

Hydrographic profiles and water samples were collected with a standard Conductivity, Temperature, Depth (CTD) rosette system with Niskin bottles. Nutrient analyses were carried out at sea using a Seal Analytical continuous-flow AutoAnalyzer 3, consistent with the methods described in the GO-SHIP repeat hydrography manual[56].

### *Sargassum* collection and identification

*Sargassum* spp. samples were collected with a dip net and sorted into the species and morphotypes, *S. natans I* and *S. fluitans III* per Parr[30]. Recent literature has indicated the increasing presence of a previously rare form *S. natans V III* on the basis of both morphology[26] and genetics[57,58]. Based on morphological similarity, any *S. natans V III* in our samples would have been classified as *S. fluitans III*.

### *Sargassum* elemental analysis and isotopic composition

Samples for each morphotype were separated into up to three replicates as quantities allowed (6 to 10 thalli/species), rinsed briefly (3 to 5 s) in deionized water, cleaned of macroscopic epizoa and epiphytes, dried in a laboratory oven at 65 to 70 °C for 48 h, and powdered with a mortar and pestle[53]. The dried *Sargassum* samples were

split in half and stored in plastic screw top vials. One half was used for arsenic analysis (see below), and the other half was shipped to the University of Georgia's Center for Applied Isotope Studies Stable Isotope Ecology Laboratory (UGA-SIEL; https://cais.uga.edu/facilities/stable-isotope-ecology-laboratory/) for analysis of $\delta^{15}N$ as well as %C and %N on a Thermo Delta V IRMS coupled to a Carlo Erba NA1500 CHN-Combustion Analyzer via a Thermo Conflo Interface. National Institute of Standards and Technology reference materials 8549, 8558, 8568, and 8569 were used to routinely calibrate working standards prepared in the laboratory. QA/QC results were incorporated into the raw data reports received by UGA-SIEL. The other part of this half sample was analyzed at UGA-SIEL for %P, where approximately 2 mg of dried tissue was weighed into crucibles, ashed at 500 °C for four hours, and extracted with 0.2 mL of Aqua Regia acid[59,60]. The acid extracts were then diluted 41:1 with deionized water for TP (as $PO_4$-P) analysis on an Alpkem 300 series analyzer.

### Sargassum arsenic content

Arsenic content of *Sargassum* tissue was measured by the University of Missouri Soil and Plant Testing Laboratory (MU SPTL) using an Inductively Coupled Plasma Optical Emission Spectrometer (ICP-OES; Agilent 5800; $\lambda_{As} = 188.980$ nm). Subsamples of rinsed, dried, and powdered *Sargassum* tissue from the GO-SHIP A20/A22 expeditions were digested at MU SPTL in 2022 according to EPA method 3052. In brief, the powdered samples were digested with a combination of $HNO_3$ acid at 175 °C for 15 min using a microwave-accelerated digestion system (CEM MARS Xpress). Samples collected prior to the 2021 surveys were retrieved from the B. Lapointe archives (1980–2018) at the Harbor Branch Oceanographic Institute, Florida Atlantic University. Prior to sending to the MU SPTL for ICP-OES analysis, this subset was digested at the National High Magnetic Field Laboratory at Florida State University (P. Morton) in 2020 following a two-step process. First, ~0.1 g aliquots of the powdered sample were carefully weighed into 15-mL Teflon digestion beakers (Savillex), to which 3 mL of concentrated $HNO_3$ acid (Fisher Optima) were added. The beakers were tightly capped and left overnight (~12 h) on a hotplate at 150 °C. The beakers were then uncapped, and the digested sample was taken to dryness (150 °C, 2–4 h). The sample residue was then digested a second time (capped, 150 °C, overnight) using 3 mL concentrated $HNO_3$ (Fisher Optima) and 200 µL of concentrated HF (Fisher Optima). The samples were taken to dryness again (uncapped, 150 °C), and the residue dissolved in 3.0 mL of 0.32 M $HNO_3$ (Fisher Optima).

All samples were analyzed at MU SPTL with ICAP-OES at a 1/100 dilution to bring the As concentrations into a working range of the matrix-matched (0.16 M $HNO_3$) external standard calibration curve. Triplicate independent digestions and analyses of four *Sargassum* tissue samples were used to determine the representative reproducibility of the sample processing and instrumental analysis methods. For more details see https://extension.missouri.edu/programs/soil-and-plant-testing-laboratory/spl-researchers.

### Backtracking of source waters

At each station where *Sargassum* was collected, the source waters were assessed by tracking particles backward in time for 60 days using a Lagrangian algorithm[61]. Surface currents were specified from the OSCAR 1/3° resolution analysis, described at https://www.esr.org/research/oscar/oscar-surface-currents/ and available at https://podaac.jpl.nasa.gov/dataset/OSCAR_L4_OC_third-deg. A total of 100 particles were released at each location where *Sargassum* was found, with random walk diffusion applied each time step $\Delta t$ with gaussian perturbations $\sigma$ defined by $\sigma^2 = 4D\Delta t$. The horizontal diffusivity $D$ was chosen to be 4000 $m^2 s^{-1}$ based on estimates for this region derived from Argo float observations[62]. In addition to being affected by surface currents, wind also influences *Sargassum* transport[31,32]. Windage factors ranging from 0.5% and 3% produce the most accurate simulations

of *Sargassum* trajectories[63,64], and a mid-point value of 2% was used here. Wind forcing was specified using daily NCEP/NCAR reanalysis[65] and validated with shipboard meteorological measurements made on R/V *Thomas G. Thompson* during voyages TN389 and TN390.

Trajectories for each virtual particle were plotted in Fig. 3 using R and TheSource[61]. Transparency was used to visually indicate relative variance in particle trajectories, and the mean path for each station was overlaid. The script used to run the model and generate the plot is available at https://gist.github.com/tbrycekelly/830f5278269ba46df15f883c1c3cccc5.

### Satellite observations

Pelagic *Sargassum* distributions (https://optics.marine.usf.edu/projects/saws.html) were derived from MODIS measurements using a floating algal index[10,66,67]. MODIS data were obtained from the U.S. National Aeronautics and Space Administration (NASA) Goddard Space Flight Center (https://oceancolor.gsfc.nasa.gov). Surface salinity distributions were obtained from SMOS Earth Explorer mission and the data were accessed from https://www.catds.fr/Products/Products-over-Ocean/CEC-Lops-SSS-SMOS-SMAP-OI-L4.

## Data availability

The *Sargassum* tissue data generated in this study have been deposited in the Figshare database under accession code https://doi.org/10.6084/m9.figshare.22684507[68]. Shipboard hydrographic data: https://cchdo.ucsd.edu/cruise/325020210316 (TN389[54]). https://cchdo.ucsd.edu/cruise/325020210420 (TN390[55]). Surface currents used for particle tracking: https://podaac.jpl.nasa.gov/dataset/OSCAR_L4_OC_third-deg[69]. Satellite-based surface salinity: https://www.catds.fr/Products/Products-over-Ocean/CEC-Lops-SSS-SMOS-SMAP-OI-L4[70]. Ocean color data for computation of satellite-based *Sargassum* distributions: https://oceancolor.gsfc.nasa.gov[71].

## Code availability

Particle tracking code: https://doi.org/10.5281/zenodo.3468524[61].

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

## Acknowledgements
*Sargassum* samples of opportunity were collected on U.S. GO-SHIP lines A20 (Ryan Woosley, Chief Scientist), and A22 (Viviane Menezes, Chief Scientist), carried out on the R/V *Thomas G. Thompson* (voyages TN389 and TN390) with the support of NSF and NOAA. We greatly appreciate the efforts of all who participated in the collection, processing, and preservation of the samples on these cruises, particularly Christine Klimkowski, Jennifer Nomura, Elizabeth Ricci, and Stephen Jalickee. Likewise, we thank Kevin Tyre, Allyson DiMarco, Dave Milmore, Jennifer Cannizzaro, and Kristie Dick for their efforts processing cruise samples. D.J.M. gratefully acknowledges partial support of this effort by the National Science Foundation (Grant Number OCE-1840381) and the National Institute of Environmental Health Sciences (Grant Number 1P01ES028938) through the Woods Hole Center for Oceans and Human Health, as well as internal support provided by the Woods Hole Oceanographic Institution and the Isham Family Charitable Fund. D.J.M. thanks O. Kosnyrev for her skillful data analysis and visualization. C.H., R.A.B., and B.E.L. acknowledge the support of the National Aeronautics and Space Administration (Grant Number 80NSSC20M0264). We thank Yingjun Zhang for the SMOS salinity analysis that went into Fig. 1. A portion of P. Morton's work was performed at the National High Magnetic Field Laboratory, which is supported by National Science Foundation Cooperative Agreement No. DMR-1644779 and the State of Florida.

## Author contributions
D.J.M. Conceptualization, methodology, formal analysis, investigation, writing-original draft, writing-reviewing and editing. R.A.B. Data curation, writing - review and editing. C.H. Investigation, writing - review and editing. T.B.K. Investigation, writing - review and editing. P.L.M. Investigation, resources, writing – review and editing. A.R.S. Formal analysis, writing – review and editing. B.E.L. Conceptualization, methodology, investigation, writing-reviewing and editing.

## Competing interests
The authors declare no competing interests.
