## [Peer Review File · Nature Communications]

Nutrient and arsenic biogeochemistry of Sargassum in the western AtlanticReviewer #1 (Remarks to the Author):

McGillicuddy, Jr., D.J., P.L. Morton, R. A. Brewton, C. Hu, T. B. Kelly, A. R. Solow, B. E. Lapointe. 2023. Nutrient and arsenic biogeochemistry of Sargassum in the western Atlantic. Review for Nature Communications

General

This paper could be recommended for publication if the nutrient fingerprinting could be extended more definitively to the central Atlantic Sargassum. The abstract for this paper is excellent in terms of defining the status of the nutrient problem and intriguing for the promise of a defining fingerprint for source nutrients in the central Atlantic that support the excessive Sargassum growth there. Data presented is used to point out differences in Sargassum nutrient content between the north Atlantic-Sargasso Sea region and the western central Atlantic. In the results section as pointed out below, it is sometimes difficult for the reader to understand which data points are being referred to in survey transects A20. It would enhance the paper to make this clear. There is a comparatively long explanation of how Arsenic and As:P indicates P limitation which may not be necessary to the paper? It is disappointing that the Arsenic fingerprinting presented, applies mostly to the SS and does not apparently apply to the new GASB where the source of nutrients needs to be defined. The authors suggests that a more comprehensive sampling of the entire breadth of the GASB might provide such a fingerprint although not based on the Arsenic which the paper emphasizes. The scale of such an effort is daunting to the authors.

Specific

P2 Line 83 omit beginning of sentence: North of 24°N, as redundant; is the first A22 supposed to be A20 in this sentence?

P4 line 139 one station..., Sentence 143 not sure what is meant by this sentence Or maybe I am looking at the wrong point? Please designate location.

P4 Line 164 Do not see Arsenic enrichment in southern most samples (or sample?) on A20? It may be useful to show limit of subtropical gyre on figures.

P4 Line 164-165 Arsenic in fluitans and natans appears somewhat different on A20 in the subtropical gyre?

P4 Line 172- 216 not sure what this material adds to the already stated P limitation in paragraph above, lines 166-171?

P5 line 201 arsenic content as a diagnostic of phosphorus limitation, what does this mean for nutrient sources with respect to the subtropical gyre? Deep nutri-cline, lack of Saharan dust or an isolated water mass??

P6 Line 234 if There is an unimportant if and no useful fingerprint for the central Atlantic in the conclusion. Sargassum in the subtropical gyre and the SS has perhaps, always been high in Arsenic and P limited (Fig 4) but biomass is and always has been low in the subtropical gyre and SS? So historically and in the future not much Sargassum with high Arsenic washes up on the beaches. The Fig 2 data show a lack of evidence that Sargassum is P limited or Arsenic is high in the GASB where Sargassum biomass is now very high?? So Arsenic fingerprinting is not an indicator for nutrient sources for the central Atlantic problem location.

Reviewer #2 (Remarks to the Author):

Key results

This study examines the carbon, nitrogen, and phosphorus content, N stable isotopic signatures, as well as arsenic content of Sargassum spp. along two latitudinal gradients from the Sargasso Sea to the Caribbean and West Tropical Atlantic (WTA). The methods of using tissue stoichiometry and N isotopes to identify potential limitation and sources of nutrients are not new, however, the

use of the As/P method is more interesting and novel. Overall, the data from the intensive sampling of Sargassum along two latitudinal transects is quite compelling, as it shows the regional differences in nutrient limitation and sources of nutrients across these regions. These data suggest nutrient levels are higher in the Northern Sargasso Sea and the Caribbean and WTA, with higher nutrient limitation in the Sargasso Sea. The relationship between As/P and % P also strongly confirms phosphorous limitation in certain regions. Although the $\delta^{15}\text{N}$ data do show variation across regions, I find the linking of these differences to riverine sources only speculative. There is no evidence that the higher signatures are due to riverine sources and that they are reaching the open ocean. Higher values could also be due to upwelling. Additionally the negative signatures in the Sargasso Sea could be due to N-fixation.

The manuscript is generally well-written and easy to follow. The methods are well described and references relevant. Overall, I think this study is significant to the field to start to understand what nutrients are important to sustain the largest algal bloom worldwide.

Specific comments:

Line 96-check reference to fig. 2. C:N data are not represented in this figure. Perhaps you mean Extended Data Fig. 1?

Line 126-what does the riverine source have to do with Fig. 3? Please explain better.

Line 132-136- This is highly speculative, although feasible, it seems like there could be other sources as well, such as upwelling.

Line 138-I do not follow the connection with the Amazon River plume and Fig. 1.

Fig. 1- It is difficult to see velocity vectors in Fig 1 a. Also do you talk about them in the text?

Fig. 2 What do the 2000 and 3000 stand for in the graphs? Include in figure legend.

Fig. 3 please add A22 and A20 to the graph

I would suggest reduction of figures in the Extended data section as a few are redundant. I would also recommend more thorough descriptions of some of the figure legends. Are you displaying means +/- standard deviation or standard error?

Extended Table 1-(mean, standard deviation?)

Extended data Fig. 1- redundant to Table 1. I would use one or the other.

Reviewer #3 (Remarks to the Author):

Review of the article "Nutrient and arsenic biogeochemistry of Sargassum in the western Atlantic". This is a very interesting study that highly contributes to the biological and physiological knowledge of Sargassum. The inundations of Sargassum on Caribbean and Florida is causing several negative ecological, social and economic impacts, hence knowledge of the species biology is urgently needed. The authors make a great work collecting many biological samples along the species distribution, as well as seawater samples. The study showed that tissue samples from Caribbean waters showed greater nitrogen (N) and phosphate (P) concentrations than those from the Sargasso Sea, which is positively correlated with nutrient seawater concentrations. These results suggest that invasion of this species might be highly related with eutrophication. Moreover, and the most interested part of the study, is the relation found between P and Arsenic content. They suggested that uptake of arsenic might occurred under P limitations. Higher arsenic content in the tissue make the species very toxic, and therefore is inedible. Overall the manuscript is very well written, and very easy to follow. I surely suggest the paper acceptance after few minors' changes.

Extended figures:

Please check your axis; it will be better if you use the same amount of decimals in all your graphs.

Also, in the extended figure 4, you should keep the same maximum value in your Y axis, as you are comparing your stations results. This will highlight the differences between stations.

Response to reviews

We are grateful for the constructive comments provided by the three referees. We have incorporated nearly all of the reviewers' suggestions into the revised version. A detailed narrative of our responses to each of their comments (in bold) follows. We note that additional changes have been made to bring the manuscript into compliance with *Nature Communications* formatting policies. Specifically, we (1) shortened the abstract to 150 words and removed references therein; (2) added a final paragraph to the introduction containing a brief summary of the major results and conclusions of the study; (3) moved the Methods and Data Availability sections to just after the Discussion; and (4) made a number of other miscellaneous changes in accordance with the guidelines. For convenience, we have provided both a "clean" and a change-tracked version of the revised manuscript.

Reviewer #1

General

This paper could be recommended for publication if the nutrient fingerprinting could be extended more definitively to the central Atlantic Sargassum. The abstract for this paper is excellent in terms of defining the status of the nutrient problem and intriguing for the promise of a defining fingerprint for source nutrients in the central Atlantic that support the excessive Sargassum growth there. Data presented is used to point out differences in Sargassum nutrient content between the north Atlantic-Sargasso Sea region and the western central Atlantic. In the results section as pointed out below, it is sometimes difficult for the reader to understand which data points are being referred to in survey transects A20. It would enhance the paper to make this clear. There is a comparatively long explanation of how Arsenic and As:P indicates P limitation which may not be necessary to the paper? It is disappointing that the Arsenic fingerprinting presented, applies mostly to the SS and does not apparently apply to the new GASB where the source of nutrients needs to be defined. The authors suggests that a more comprehensive sampling of the entire breadth of the GASB might provide such a fingerprint although not based on the Arsenic which the paper emphasizes. The scale of such an effort is daunting to the authors.

We thank the referee for this positive assessment and constructive suggestions. We have modified the results section to include geographic coordinates of the specific stations that are called out in the text. We have clarified the fact that the fingerprinting idea pertains to nitrogen and phosphorus, not to arsenic—which is an indicator of phosphorus stress that does not depend on the source. We stand by the novelty of the arsenic biogeochemistry presented herein, consisting of a theoretical prediction tested with our observations—which has important implications for management if the nutrient source(s) turn out to be anthropogenic. Details of the revisions are described below.

Specific

P2 Line 83 omit beginning of sentence: North of 24oN, as redundant; is the first A22 supposed to be A20 in this sentence?

Fixed.

P4 line 139 one station..., Sentence 143 not sure what is meant by this sentence Or maybe I am looking at the wrong point? Please designate location.

Station position now indicated in the text: 17° N, 67° W.

P4 Line 164 Do not see Arsenic enrichment in southern most samples (or sample?) on A20? It may be useful to show limit of subtropical gyre on figures.

We have clarified that the lowest Arsenic concentrations are in the GASB (Caribbean and Western Tropical Atlantic samples), and the modest enrichment on A20 to which we refer is the southernmost station at 9° N, 52°W. We have also noted modest enrichment of Arsenic in the Northern Sargasso Sea samples on A20 (31-33° N, 52°W).

P4 Line 164-165 Arsenic in *fluitans* and *natans* appears somewhat different on A20 in the subtropical gyre?

We have rephrased the statement to acknowledge interspecies variability, and point to Supplementary Table 1 to support the statement that there are no systematic differences:

“There are no systematic differences in arsenic content between *S. fluitans* and *S. natans* (Supplementary Table 1), although interspecies variability is apparent at some stations (Fig. 2).”

P4 Line 172- 216 not sure what this material adds to the already stated P limitation in paragraph above, lines 166-171?

This material develops a theoretical prediction for how the As:P ratio varies as a function of P content, which to our knowledge has not been described before. Moreover, we test the theoretical prediction with our data in Figure 4. This is a truly novel aspect of the paper and thus have chosen to keep this material intact.

P5 line 201 arsenic content as a diagnostic of phosphorus limitation, what does this mean for nutrient sources with respect to the subtropical gyre? Deep nutri-cline, lack of Saharan dust or an isolated water mass??

This issue is discussed in the second paragraph of this section:

“The high arsenic content of Sargassum in the subtropical gyre is accompanied by the lowest phosphorus content, which is consistent with the depression of the phosphocline (Supplementary Fig. 3) and surface dissolved phosphate concentrations generally below the limit of detection (Supplementary Fig. 4). Thus, the arsenic to phosphorus ratio of Sargassum of the subtropical gyre stands out as uniquely high in that region (Fig. 2), where Sargassum is known to be phosphorus limited⁴².”

P6 Line 234 if There is an unimportant if and no useful fingerprint for the central Atlantic in the conclusion. Sargassum in the subtropical gyre and the SS has perhaps, always been high in Arsenic and P limited (Fig 4) but biomass is and always has been low in the subtropical gyre and SS? So historically and in the future not much Sargassum with high Arsenic washes up on the beaches. The Fig 2 data show a lack of evidence that Sargassum is P limited or Arsenic is high in the GASB where Sargassum biomass is now very high?? So Arsenic fingerprinting is not an indicator for nutrient sources for the central Atlantic problem location.

We agree with the referee that arsenic is not an indicator of nutrient source(s)—arsenic content is an indicator of phosphorus limitation, regardless of what the phosphorus source(s) may be. The fingerprinting of the nutrient sources we suggest has to do with nitrogen and phosphorus, not arsenic. We have clarified in both the abstract and in the conclusions that fingerprinting pertains to nitrogen and phosphorus content.

The referee is correct that “*Sargassum* in the subtropical gyre and the SS has perhaps, always been high in Arsenic and P limited (Fig 4) but biomass is and always has been low in the subtropical gyre and SS...” As for phosphorus limitation of Sargasso Sea populations, the evidence for that is presented in the second paragraph of the section on Arsenic Biogeochemistry (see above) which references both the present data (Supplementary Figs. 3 and 4) as well as a prior study [ref. 42, Lapointe (1986)].

The referee is also correct that arsenic is relatively low in the GASB where biomass is high, and we have emphasized that point on the new line 161: “Arsenic content was lowest in the GASB (Caribbean and Western Tropical Atlantic samples) and highest in the Sargasso Sea, particularly in the eastern transect A20.”

The main point of the sentence starting on line 234 is that a reduction in phosphorus supply to the GASB (*relative to nitrogen*) could increase the arsenic content—which would have important implications for management purposes in case the nutrient sources turn out to be anthropogenic. We have clarified the sentence accordingly and added prose with the additional point concerning management.

Reviewer #2:

Key results

This study examines the carbon, nitrogen, and phosphorus content, N stable isotopic signatures, as well as arsenic content of *Sargassum* spp. along two latitudinal gradients from the Sargasso Sea to the Caribbean and West Tropical Atlantic (WTA). The methods of using tissue stoichiometry and N isotopes to identify potential limitation and sources of nutrients are not new, however, the use of the As/P method is more interesting and novel. Overall, the data from the intensive sampling of *Sargassum* along two latitudinal transects is quite compelling, as it shows the regional differences in nutrient limitation and sources of nutrients across these regions. These data suggest nutrient levels are higher in the Northern Sargasso Sea and the Caribbean and WTA, with higher nutrient limitation in the Sargasso Sea. The relationship between As/P and % P also strongly confirms phosphorous limitation in certain regions. Although the $\delta^{15}\text{N}$ data do show variation across regions, I find the linking of these differences to riverine sources only

speculative. There is no evidence that the higher signatures are due to riverine sources and that they are reaching the open ocean. Higher values could also be due to upwelling. Additionally the negative signatures in the Sargasso Sea could be due to N-fixation.

The manuscript is generally well-written and easy to follow. The methods are well described and references relevant. Overall, I think this study is significant to the field to start to understand what nutrients are important to sustain the largest algal bloom worldwide.

We thank the referee for this positive assessment. As per the referee's wise advice, we have bolstered the discussion of $\delta^{15}\text{N}$ and nitrogen sources to include upwelling and nitrogen fixation, adding relevant references.

Specific comments:

Line 96-check reference to fig. 2. C:N data are not represented in this figure. Perhaps you mean Extended Data Fig. 1?

The sentence is correct as is. The C:N ratio is presented in Supplementary Fig. 2 referenced in the preceding sentence; the C and N content used to explain that are shown in Fig. 2.

Line 126-what does the riverine source have to do with Fig. 3? Please explain better.

The figure reference has been clarified: "as indicated by spatial connectivity depicted in Fig. 3."

Line 132-136- This is highly speculative, although feasible, it seems like there could be other sources as well, such as upwelling.

Indeed, we have added a sentence stating upwelling as another possible source: "Alternatively, enrichment of $\delta^{15}\text{N}$ in samples from the northern Sargasso Sea could reflect upwelling and/or vertical mixing, as values of 2 ‰ are characteristic of nitrate in the upper thermocline in that region."

Line 138-I do not follow the connection with the Amazon River plume and Fig. 1.

The figure reference has been revised to "as evidenced by the salinity distribution in Fig. 1."

Fig. 1- It is difficult to see velocity vectors in Fig 1 a. Also do you talk about them in the text?

The vectors are now plotted in black instead of light blue to improve clarity. Yes, they are referred to in the text (e.g., Gulf Stream, eddies, etc.).

Fig. 2 What do the 2000 and 3000 stand for in the graphs? Include in figure legend.

These are the 2000 m and 3000 m isobaths, and that info has been added to the legend of Fig. 2 and Fig. 1.

Fig. 3 please add A22 and A20 to the graph

This has been done.

I would suggest reduction of figures in the Extended data section as a few are redundant. I would also recommend more thorough descriptions of some of the figure legends. Are you displaying means +/- standard deviation or standard error?

We could not identify any redundancies other than those noted below. The +/- values are 95% confidence intervals, and that information has been added to the relevant captions. Additional information has been added to the figure captions, including identification of the 2000 m and 3000 m isobaths in Supplementary Fig. 2.

Extended Table 1-(mean, standard deviation?)

These are the means and 95% confidence intervals, and that information has been added to the caption for Supplementary Table 1 and Supplementary Fig. 1.

Extended data Fig. 1- redundant to Table 1. I would use one or the other.

While there is some overlap in the information, Supplementary Fig. 1 actually has more quantities plotted than are reported in Supplementary Table 1, specifically As:C, As:N, and As:P. Moreover, we feel that both presentations are valuable, with the figure providing graphical comparison among the quantities and the table providing the actual numbers which are helpful for reference. For both of these reasons we have chosen to retain both the figure and the table.

Reviewer #3 (Remarks to the Author):

Review of the article “Nutrient and arsenic biogeochemistry of Sargassum in the western Atlantic”. This is a very interesting study that highly contributes to the biological and physiological knowledge of Sargassum. The inundations of Sargassum on Caribbean and Florida is causing several negative ecological, social and economic impacts, hence knowledge of the species biology is urgently needed. The authors make a great work collecting many biological samples along the species distribution, as well as seawater samples. The study showed that tissue samples from Caribbean waters showed greater nitrogen (N) and phosphate (P) concentrations than those from the Sargasso Sea, which is positively correlated with nutrient seawater concentrations. These results suggest that invasion of this species might be highly related with eutrophication. Moreover, and the most interested part of the study, is the relation found between P and Arsenic content. They suggested that uptake of arsenic might occurred under P limitations. Higher arsenic content in the tissue make the species very toxic, and therefore is inedible.

Overall the manuscript is very well written, and very easy to follow. I surely suggest the paper acceptance after few minors' changes.

We thank the referee for this positive assessment.

Extended figures:

Please check your axis; it will be better if you use the same amount of decimals in all your graphs. Also, in the extended figure 4, you should keep the same maximum value in your Y axis, as you are comparing your stations results. This will highlight the differences between stations.

As per the referee's suggestion, we have revised Supplementary Fig. 4 so that the y-axis limits are the same in both columns.

As for the decimal places, it is not practical to use the same number in all cases given the wide variation in precision of the measurements, from arsenic content to temperature. In all cases we have chosen an appropriate number of significant figures to report, keeping them consistent among the plots to facilitate comparison.